

# Membership inference attack on differentially private block coordinate descent

Shazia Riaz[1,2], Saqib Ali[2,3], Guojun Wang[3], Muhammad Ahsan Latif[2] and Muhammad Zafar Iqbal[4]

[1] School of Computing, Macquarie University, Sydney, Australia
[2] Department of Computer Science, University of Agriculture, Faisalabad, Punjab, Pakistan
[3] School of Computing, Guangzhou University, Guangzhou, China
[4] Department of Mathematics and Statistics, University of Agriculture Faisalabad, Faisalabad, Punjab, Pakistan

## ABSTRACT

The extraordinary success of deep learning is made possible due to the availability of crowd-sourced large-scale training datasets. Mostly, these datasets contain personal and confidential information, thus, have great potential of being misused, raising privacy concerns. Consequently, privacy-preserving deep learning has become a primary research interest nowadays. One of the prominent approaches adopted to prevent the leakage of sensitive information about the training data is by implementing differential privacy during training for their differentially private training, which aims to preserve the privacy of deep learning models. Though these models are claimed to be a safeguard against privacy attacks targeting sensitive information, however, least amount of work is found in the literature to practically evaluate their capability by performing a sophisticated attack model on them. Recently, DP-BCD is proposed as an alternative to state-of-the-art DP-SGD, to preserve the privacy of deep-learning models, having low privacy cost and fast convergence speed with highly accurate prediction results. To check its practical capability, in this article, we analytically evaluate the impact of a sophisticated privacy attack called the membership inference attack against it in both black box as well as white box settings. More precisely, we inspect how much information can be inferred from a differentially private deep model's training data. We evaluate our experiments on benchmark datasets using AUC, attacker advantage, precision, recall, and F1-score performance metrics. The experimental results exhibit that DP-BCD keeps its promise to preserve privacy against strong adversaries while providing acceptable model utility compared to state-of-the-art techniques.

# INTRODUCTION

Privacy has now become the primary concern of all domains using any form of deep learning, *e.g.*, computer vision (*Ruan et al., 2020*), natural language processing (NLP) (*Masri & Al-Jabi, 2023*), medical diagnosis (*Sharma et al., 2022*), web search (*Khan, Mohibullah & Islam, 2017*), *etc.* Since a deep learning (DL) model is trained on a massive

Corresponding authors
Saqib Ali, saqib@uaf.edu.pk
Guojun Wang, csgjwang@gzhu.edu.cn

amount of data, usually crowdsourced from a population (especially, large internet companies such as Facebook, Twitter, Google, Apple, Amazon, BigML, *etc.*, have a large user base to collect data for training their DL models). The training data may contain sensitive information about individuals, who have no idea how their data will be used after collection. Normally, a DL model has enough capacity to memorize the training data, thus, has a likelihood of leaking the information (*Leino & Fredrikson, 2020*). An adversary queries the DL model to infer private information about the individual samples of the training data by exploiting the prediction vector, thus creating severe privacy concerns.

From the privacy perspective, the most prominent privacy issue is the membership inference attack (MIA) that infers whether a particular sample is included in the training dataset of the targeted model (*Shokri et al., 2017*). The other famous attacks are model inversion attack that aims to recover a sample from the model's training dataset based on guessing made on the prediction vector outputted by the model, property inference attack to comprehend the attributes of the model's training dataset, model extraction attack to extract a trained model exploiting its predictions, and generative adversarial network (GAN) based attacks to synthesize instances from a real dataset (*Rigaki & Garcia, 2020*).

Along with developing new attacks, the research community geared up to focus their attention on privacy-preserving deep learning (PPDL). PPDL's main objective is to prevent an adversary from inferring sensitive information about the model's training data. The work done in this domain uses different mechanisms such as differential privacy (DP) (*Dwork & Roth, 2014*; *Sei, Okumura & Ohsuga, 2016*; *Oneto, Ridella & Anguita, 2017*), homomorphic encryption (*Fontaine & Galand, 2007*; *Xie et al., 2014*), partial parameters sharing (*Phong et al., 2017*), functional exponentiation (*Zhang et al., 2012*), model splitting (*Dong et al., 2018*), *etc.* Usually, most PPDL techniques do not allow trained model publishing to achieve privacy. However, DP-based techniques claim to be privacy preserved even by sharing these trained models publicly. Since these DP-based techniques carry out differentially private model training and provide a guarantee to protect against a strong adversary despite its background knowledge and computational power. As a result, DP has become the most prominent and widespread mechanism used for guaranteeing privacy in deep learning. Therefore, many big corporations, *i.e.*, Google, Apple, and US Census Bureau, have employed it.

As differential privacy has shifted from theory to practice, it has motivated researchers to pay attention to the optimization and evaluation of differentially private deep learning (DPDL) algorithms. Among these existing algorithms, differentially private stochastic gradient descent (DP-SGD) (*Abadi et al., 2016*) is the extensively used optimization algorithm in deep learning for classification problems, *e.g.*, convolutional neural networks and feedforward neural networks. Therefore, it has become part of the TensorFlow privacy library due to its widespread use. Another recent work by *Riaz et al. (2023)* presents differentially private block coordinate descent (DP-BCD), which claims to provide a modest privacy guarantee while maintaining utility at an acceptable tradeoff compared to other state-of-the-art techniques.

Despite the efforts spent on achieving DP through various mechanisms, there needs to be an appropriate method to evaluate the robustness of the DPDL models. Since privacy

analysis provides the details of privacy budget ($\varepsilon$) consumption along with the model's accuracy on prediction guaranteeing privacy while providing utility. However, the promised privacy preservation guarantee cannot provide the extent to which the DPDL model is resilient to the above-discussed attacks and which privacy risks with high certainty can be addressed by the model. One method to resolve this issue is to generate a sophisticated attack on the DPDL technique to check its robustness against the targeted attack. Among all existing attacks, MIA is claimed to be an effective attack since it infers the presence of a particular data sample in the training dataset of a DL model, representing an explicit privacy breach. Therefore, MIA has attracted researchers, and several enhanced versions have been designed since its initial launch (*Truex et al., 2019*; *Jayaraman et al., 2021*; *Maini, Yaghini & Papernot, 2021*; *Choquette-Choo et al., 2021*; *Hu et al., 2022*). Moreover, it is applicable to almost all types of datasets, such as images, genomic data, relational data, and others (*Shokri et al., 2017*; *Long et al., 2018*; *Salem et al., 2019*), for the privacy test when sharing a trained DL model. Thus, researchers have adopted it to evaluate the privacy endorsed by the famous DP-SGD. In the wake of such sophisticated attack models, it becomes unavoidable to ask every DPDL model how well its privacy guarantee withstands the MIA to prove its effectiveness.

In this article, we try to answer this question and evaluate the privacy guarantee of DP-BCD by launching MIA against it. We conduct the attack in the white box setting, where an adversary can access the model's internals, such as model type, architecture, *etc*. For this, we train the attack model on a dataset created by collecting the prediction results of the shadow model. The shadow model's structure is similar to the corresponding target model in white box settings. The proposed mechanism is also checked by performing MIA in black box settings without shadow model training, where an adversary can access only the model predictions by querying the model. We measure the success of MIA on DP-BCD in terms of precision, recall, attacker advantage, and AUC (area under the ROC curve) as performance metrics. The empirical results demonstrate that compared to DP-SGD, DP-BCD is less vulnerable to MIA while offering high utility. In some cases, membership inference risks of DP-SGD resemble a non-private DL model. Since the capability of DP is always questioned for not providing an intuitive guide for selecting privacy budget parameter $\varepsilon$. Consequently, DPDL applications tend to select a random value of $\varepsilon$ to maintain a tradeoff between privacy and the utility of the model. Therefore, this study elaborates on choosing a suitable amount of privacy budget $\varepsilon$ and how it practically affects privacy.

In summary, our contributions are as follows.

- We evaluate membership privacy threat by implementing MIA in white box settings with only one shadow model on the DP-BCD mechanism for heavy noise ($\varepsilon = 0.5, \delta = 10^{-3}$) and moderate noise ($\varepsilon = 1, \delta = 10^{-4}$).
- We also implement MIA without shadow model training in black box settings, ultimately arriving at a model and data-independent adversary. This type of MIA is also performed for two different choices of privacy cost, *i.e.*, heavy noise ($\varepsilon = 0.5, \delta = 10^{-3}$) and moderate noise ($\varepsilon = 1, \delta = 10^{-4}$).

- Extensive experiments demonstrate that severe MIA in both white and black box settings with a suggested privacy budget in the form of moderate and heavy noise levels prove the robustness of the DP-BCD model compared to other state-of-the-art techniques.

The rest of the article is organized as follows. "Background" describes the background of privacy issues in deep learning and various attacks. The detailed methodology of designing membership inference attacks on the differentially private deep learning model is presented in "Methodology". In "Experimental Evaluation and Results", we give details about experimental evaluation and results. Finally, we conclude by mentioning the possible future scope of this work in "Conclusion".

## BACKGROUND

In this section, the literature on privacy issues in deep learning and then attacks on differentially private models is discussed in detail.

### Privacy issues in deep learning

DL is a state-of-art artificial intelligence (AI) mechanism that significantly improves prediction accuracy using powerful data abstraction capabilities on highly-structured and large-scale datasets (*LeCun, Bengio & Hinton, 2015*). The trained DL model has the potential to leak the sensitive information of individuals learned from the data it was trained on. For example, *Homer et al. (2008)* inferred the existence of a specific genome in the genomic training dataset of a trained model by exploiting the published statistics of genotype count distributions. In *Calandrino et al. (2011)*, background information about a customer and changes in the public output of the collaborative recommender system was used to execute an inference attack that infers the customer's transaction that causes the changes in the system's output.

Recent trends such as ML-as-a-Service (MLaaS) use a black-box API to provide classification service. These APIs use the features of input samples containing sensitive information. The adversary exploits the overfitting of these models to extract sensitive information since the model's internal wirings implicitly remember some details about training data. White-box settings make ML models more vulnerable to attacks like training data extraction attacks. For example, *Ateniese et al. (2015)* trained an attack model that extracts meaningful statistics about the training dataset of target ML models (*i.e.*, SVM and HMM) by exploiting the information about their model parameters.

Moreover, the adversary uses the structure and type of the model, which turns the ML against itself. The privacy attacks typically target DL models during their training and inference phases. One sophisticated attack of this type is the model inversion attack (*Fredrikson et al., 2014*), in which authors try to infer the input sample's hidden features corresponding to the output. Pharmacogenetics analysis was carried out in this attack to capture the association between a drug dose and a patient's genotype. However, it is not considered an actual privacy breach due to the existence of inherent medical evidence between the components. In a more aggressive attack of this kind (*Fredrikson, Jha &*

*Ristenpart, 2015*), the attacker probes the facial recognition system by giving a randomly generated image as input and using the confidence score of the output prediction vector to refine the image. The recovered image is humanly recognizable, having 95% matching accuracy with the actual image in the training set.

One more well-known attack is the model extraction attack crafted by *Tramèr et al. (2016)* to retrieve the model parameters of a specific model trained on sensitive private training data. The attacker aimed to mimic the functionality of the target model to train an adversarial model by exploiting the close connection between the model parameters and the training dataset. A contemporary attack designed by *Sharif et al. (2016)* tries to deceive a biometric facial recognition system. Their proposed attack worked by training a deep face recognition model in which the adversary either avoids the identification as an authenticated individual or mimics another individual. Model stealing attacks (*Hitaj, Ateniese & Perez-Cruz, 2017*; *Orekondy, Schiele & Fritz, 2019*) also make the adversarial use of machine learning (ML) by designing a generative adversarial network (GAN) based attack, where a malicious participant trains a model in a collaborative environment by deceiving the honest participant to release extra information about the private training dataset. In attribute inference attacks (*Melis et al., 2019*; *Malekzadeh, Borovykh & Gündüz, 2021*) during the training phase, the adversary is always active and uses extra information about the training data samples to produce prototypical samples with the same target model's training set distribution. Collaborative model training has opened new avenues for adversaries where a malicious participant can steal information by training a GAN (*Goodfellow et al., 2020*; *Gui et al., 2023*).

Among the other attacks, there exist poisoning attacks (*Chen & Koushanfar, 2023*) which poison the model's training dataset to falsify the model predictions. The famous examples are data poisoning attacks (*Li et al., 2016*), model poisoning attacks (*Marulli, Verde & Campanile, 2021*; *Ali et al., 2023*), and label-flipping attacks (*Imam & Vassilakis, 2019*; *Zhang et al., 2021*), and others.

Most of the ML-based attacks typically suffer from various issues, *i.e.*, model inversion attacks have limited capabilities and only apply to face recognition systems. Similarly, model extraction attacks are reported to work with specific model types, whereas some attacks require a considerable amount of background information for fruitful outcomes. In strong contrast, MIA, designed by *Shokri et al. (2017)*, is an innovative and robust attack that can work with any model environment and setting with limited access to information. Given an input sample and target model, the objective of the attack is to infer the presence of the particular sample in the training dataset of the targeted model. Since the inception of MIA, a number of its variants have been proposed in the literature (*Yeom et al., 2018*; *Salem et al., 2019*; *Truex et al., 2019*; *Jayaraman et al., 2021*; *Maini, Yaghini & Papernot, 2021*; *Choquette-Choo et al., 2021*; *Hu et al., 2022*). Our attack implementation to verify the robustness of DP-BCD is based on the working of MIA and is discussed in detail in the following section.

## Differential privacy and attacks

Differential privacy is a probabilistic privacy mechanism, a strong notion of privacy, established by *Dwork et al. (2006b)*, *Dwork, Rothblum & Vadhan (2010)*, *Dwork & Roth (2014)*, and offers a more robust privacy guarantee for algorithms on aggregate datasets. It guarantees that by analyzing a DL model's predictions, it is impossible to infer the presence or absence of a particular sample in the training dataset of the model. It is independent of the adversary's computational power or background information. Specifically, implementing a differentially private mechanism M on two neighboring datasets, $D_1$ and $D_2$ (with a difference of one sample only), makes the output indistinguishable. Differential privacy (DP) is formally defined as follows.

**Definition 1:** A randomized mechanism M: D→R having domain D and range R is said to establish $(\varepsilon, \delta)$-DP if, for any two neighboring datasets $D_1$ and $D_2 \in$ D with a difference of one sample only and for any subset of possible outcomes S $\subseteq$ R, then it satisfies the following.

$$pr[M(D_1) \in S] \leq pr[M(D_2) \in S] * e^{\varepsilon} + \delta \qquad (1)$$

The parameter $\varepsilon$ denotes the amount of privacy budget required to achieve the desired privacy, which maintains the tradeoff between the accuracy of DL model predictions and privacy preserved by the differentially private mechanism. The small $\varepsilon$ value indicates a higher level of privacy with lower amount of information leakage. The additive term $\delta$ is a relaxation inserted in $\varepsilon$-DP (pure DP). $(\varepsilon, \delta)$-DP is a variant of pure $\varepsilon$-DP introduced by *Dwork et al. (2006a)*, which enables the likelihood of breaking the pure $\varepsilon$-DP with probability $\delta$, the value typically chosen as $< \frac{1}{|D|}$.

One way to achieve privacy preservation of DP is to add a calibrated noise to perturb the query function results. The amount of noise added is proportional to the function's sensitivity, sampled from Laplace or Gaussian distributions for achieving $\varepsilon$-DP or $(\varepsilon, \delta)$-DP, respectively. Sensitivity is the maximum change determined by altering a single dataset sample. It is defined as follows.

**Definition 2:** Consider any query function $f$: D→R, executed on datasets $D_1$ and $D_2 \in$ D that differ only in the single record; then, the sensitivity of $f$ denotes the maximal difference of its outputs on $D_1$ and $D_2$. Formally,

$$S_f = \max_{D_1, D_2} \| f(D_1) - f(D_2) \|_1 \qquad (2)$$

where $\| f(D_1) - f(D_2) \|_1$ is the norm of the vector.

For instance, the noise is normally drawn from the Gaussian distribution relaxing the privacy by the parameter $\delta$.
which is defined by

$$M(D) = f(D) + N(0, S_f^2 . \sigma^2) \qquad (3)$$

$N(0, S_f^2 . \sigma^2)$ is the Gaussian noise with mean 0 and variance $S_f^2 . \sigma^2$. The noise variance is proportional to the function's sensitivity. The technique of differing record sensitivity is

normally used for databases. In contrast, the DL model comprises multiple layers of neurons that perform various types of nonlinear transformations of the inputs received on the input layer. The function $f$ is the composite function composed of different operations performed at the DL model layers from inputs to outputs. Hence, one way to implement the DP in DL is to limit the sensitivity of the individual function during computations performed at each layer and inject noise to these functions having bounded sensitivity.

Various DPDL mechanisms have been presented in the literature to cope with the privacy leakage issue. These mechanisms theoretically prove their privacy preservation and provide a privacy analysis in the form of privacy cost (*Yu et al., 2019*; *Adesuyi & Kim, 2020*; *Xu et al., 2020*). However, only a few of these mechanisms are tested by attack implementations to verify their robustness against attacks. For example, *Hay et al. (2016)* introduced an innovative framework, DPBENCH, for the standardized evaluation of 15 differentially private mechanisms on 27 datasets. Similarly, a statistical approach is presented to detect the violation of standard DP in the implementation of various incorrect DP-based mechanisms, and counter-examples are suggested for the correctness of implementations (*Ding et al., 2018*). However, the impact of different variants of DP and the selection of an appropriate amount of privacy budget (ε) is not considered in this approach. *Carlini et al. (2019)* analyzed the efficiency of DP to guard against the memorization of training data by neural networks and demonstrated that DP-RMSProp removes the privacy risk by carefully selecting the clipping threshold and noise level. However, they do not suggest the appropriate values of ε to be used to achieve the best results of DP.

*Jayaraman & Evans (2019)* evaluated DP for the accuracy and privacy of ML models. They studied the effect of different ε values on available relaxed versions of DP, specifically for gradient perturbation mechanisms. *Li et al. (2013)* argued that variants of DP compromise privacy by relaxing the standard concept of DP while getting better utility. *Liu et al. (2019)* performed a thorough evaluation of DPML models using the Neyman–Pearson criterion and quantified privacy breaches in them for different choices of ε. They suggested the choice of privacy parameter ε according to the auxiliary information available to the adversary in the form of the probability distribution of the data or tuple dependencies knowledge (*Kontorovich, Sadigurschi & Stemmer, 2022*). Another attempt made by *Ali et al. (2022)* introduced the concept of enhanced DP to preserve the privacy of dependent tuples of a correlated dataset. They also performed MIA on their model to check the validity of the proposed approach and concluded that enhanced DP proved successful in preserving the privacy of the correlated dataset. However, the approach utilized a considerable amount of privacy budget to provide an acceptable utility.

There are several remarkable techniques proposed in the literature to cope with the privacy breach issues of DL models. Among all these techniques, DP-SGD, the differentially private version of SGD, is the most prominent since SGD is the most popular algorithm for ML model optimization. Therefore, it has attracted most researchers' attention for privacy analysis by attack implementations. For example, *Rahman et al. (2018)* analyzed the impact of MIA on the DP-SGD mechanism for different choices of ε. Their experimental results demonstrated that DPDL models preserve privacy at the cost of

model utility, *i.e.*, an acceptable utility made the model vulnerable to inference attacks. *Jagielski, Ullman & Oprea (2020)* evaluated the impact of data poisoning attacks on DP-SGD to investigate the extent to which it preserved privacy while providing acceptable accuracy in practice. *Bernau et al. (2019)* compare the MIA impact on local and central differential privacy. The authors calculated the performance metric AUC and deduced that both exhibit similar characteristics for privacy-utility tradeoffs for different values of ε. Recently, a sampling attack for MIA has been proposed to check the applicability of DP as a guard against privacy leakage (*Rahimian, Orekondy & Fritz, 2021*). Besides DP-SGD, another defense mechanism called knowledge distillation is also checked against different inference attacks. The authors called their proposed technique ML-Doctor and claimed that both defense techniques can not withstand all types of inference attacks effectively (*Liu et al., 2022*). Similarly, *Tang et al. (2022)* devised a Split-AI ensemble-based self-distillation approach to preserve the privacy of ML models against MIAs. The proposed approach provides an attractive tradeoff between model utility and privacy. However, it cannot provide verifiable guarantees against all adversaries as opposed to DP.

In deep learning, DP-SGD is considered a state-of-the-art privacy-preserved optimization mechanism. Therefore, its capabilities against prominent attacks are analyzed in the literature. The objective of this article is to evaluate the impact of MIA on another recently proposed differentially private optimization mechanism, *i.e.*, DP-BCD, and compare it against state-of-the-art mechanisms available in the literature to test its competencies. We adopt this approach since the target DPDL model is versatile, efficient, converge in early epochs, and provides state-of-the-art privacy and utility trade-off.

## METHODOLOGY

This section describes the detail of the MIA and the targeted differentially private deep learning technique, *i.e.*, DP-BCD (*Riaz et al., 2023*). The attack is performed in white black box and black box settings. In white box settings, model internals such as type, architecture, *etc.*, are known to the adversary. Therefore, a shadow model is trained to generate the dataset for the training of the attack model. Whereas in white box settings, no shadow model training is required; only the target model outputs are used to train the attack model (*e.g.*, losses, logits, predictions). The following subsections provide the detail of each of these techniques.

### Membership inference attacks

MIA exploits the fact that all DL models predict differently on training and testing data, *i.e.*, it reacts differently to never seen before data. The most probable factors that make the inference attack successful are, exploiting the target model's overfitting, classification problem complexity, in-class standard deviation, and type of DL model targeted (*Yeom et al., 2018*; *Truex et al., 2019*). However, an adversary mostly takes advantage of overfitting of the model as during training most DL models consisting of too many layers with each layer having a large number of neurons, store training data on its internal wirings instead of learning from it, thus, falls prey to overfitting. *Shokri et al. (2017)* were the pioneers in introducing the MIAs in 2017. Since then, it has grabbed the researchers' attention to

mitigate the vulnerabilities of a given model, quantify the membership risk influences, and research to make these attacks more efficient. The fundamental theme of MIAs is straightforward, *i.e.*, having a DL model trained on some training dataset, it determines whether a particular sample was present in the model's training dataset.

### How do MIA work?

Normally, the functionality of our MIA is analogous to the original attack model described by *Shokri et al. (2017)*. It targets a trained classification DL model $f_{target}$ that produces a prediction vector containing confidence scores for each output class when a data sample $x_i$ is fed. Since, we are targeting DP-BCD, which is a public model, means that model's internal, *i.e.*, architecture, parameters, input and output formats, are accessible to the adversary. It is also supposed that the adversary may know the population from which target model's training dataset was drawn. In addition to the background knowledge of the population, having access to the target model's input and output formats enables an adversary to retrieve samples independently from the same population.

MIA works by building a binary attack model $f_{attack}$ that takes the prediction vector produced by the target model and a sample $x_i$. The attack model then decides whether sample $x_i$ was a member of the training dataset X of the target model. The systematic flow to accomplish a membership inference attack is illustrated in Fig. 1. For the attack model $f_{attack}$, the adversary first constructs $K$ (one for each class) shadow models $f_{shadow}^j$. It is assumed that the shadow models $f_{shadow}^j$ mimic the behavior of the target model $f_{target}$ since its training data is drawn from the same population as used for the target model's training dataset X. The difference is that shadow models training dataset $X'$, and ground truth $y'$ are known to the adversary. Next, the shadow models are trained on the sampled dataset. Afterward, the prediction vector's confidence score produced by the shadow models both for train and test datasets is integrated to create input-output pairs $(x_i', f_{shadow}^j, y_i')$ for the training of the attack model in an attempt to make it learn the task of distinguishing between members and nonmembers according to the target DL model's prediction performed on them.

### Shadow models

As mentioned earlier, the attack model comprises several shadow models that are generally constructed for each output class. According to the pioneer attack model (*Shokri et al., 2017*), the larger the number of shadow models, the more accurate the results of the attack model will be. They also described three methods for generating shadow datasets, *i.e.*, noisy real-world data that resembles the original dataset X, data synthesis with the help of $f_{target}$, or using statistics over X. Since the training dataset of each of these shadow models resembles the format and distribution of the target model's private training dataset, thus, there is a chance that individual datasets for the shadow model may contain similar data samples. However, it cannot be assumed that the target model's dataset (train and test) and the shadow model's datasets overlap.

Keeping in view the complexity and training cost of shadow training models, we implement the model and data-independent MIA revised by *Salem et al. (2019)*. It notably

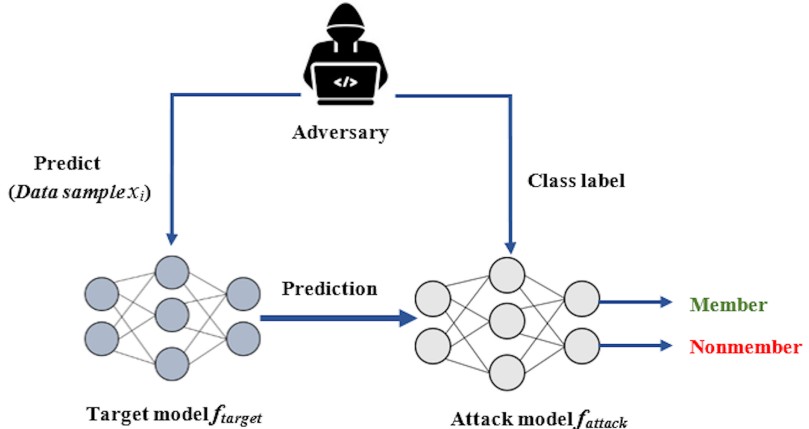

**Figure 1 The overview of membership inference attacks (MIA).**

reduces the cost of performing the attack, thereby giving the same accuracy as given by *Shokri et al. (2017)* by implementing multiple shadow models. We perform MIA in two settings, *i.e.*, attack with the shadow model training or without the shadow model training.

## Attack with the shadow model training

In this case, only one shadow model is required instead of multiple. We adopt the assumption made by *Shokri et al. (2017)* that the training data $D_{shadow}$ for the shadow model is drawn from the population as for the target model. Further, it is assumed that the shadow model follows the training procedure and type of target model. To this end, adversaries can either use the same machine learning as a service (MLaaS) as used for building the target model or approximate the behavior of the target model by performing a model extraction attack. Since a shadow model, which is built by taking the services of MLaaS, follows a pay-per-query business model, thus is a costly method. We assume that the target model is public; thus, knowing its type and architecture is easy. Therefore, the same architecture is adopted for the shadow model as for its corresponding target model. The adversary records the shadow model's output prediction vectors $V(x_i)$ generated on training and test datasets. These prediction vectors are stored in a single database, which is split into different portions corresponding to their labels for the "in" and "out" classes. For the shadow model training, the $D_{shadow}$ is split into two disjoint sets, *i.e.*, $D_{shadow}^{Train}$ and $D_{shadow}^{Out}$ for training the shadow model and as non-members, respectively.

## Attack without the shadow model training

For MIA implementation in the absence of a shadow model, the adversary only uses the original (target) model's predictions on the target data sample, which is enough to infer its membership in the target model's training dataset. It does not require any shadow model training to imitate the original model's behavior, strengthening the attack more efficiently. In this setting, the target model $f_{target}$ under attack kind of serves as a "shadow model" $f_{shadow}$ that impeccably approximates its own behavior and makes the MIA much simpler and more efficient.

### Attack model

The adversary implements the attack model as a binary meta classifier $f_{attack}$. The attack model is implemented differently for two settings of MIA, for instance, with shadow model, and without shadow model.

## Attack model with the presence of a shadow model

In this scenario, we implement the attack model as a supervised binary classifier which requires labeled training data, such as ground truth membership. To achieve this, the adversary trains the shadow model and uses its output posteriors to get the ground truth for the training of target model $f_{target}$. For instance, she employs the trained shadow model to retrieve posterior probabilities produced by the prediction performed on all data samples in $D_{shadow}$ that comprise $D_{shadow}^{Train}$ and $D_{shadow}^{out}$. She picks the three topmost posteriors (two in the case of two classes of a dataset) corresponding to each sample of $D_{shadow}$ as its prediction vector. In case, the data sample is from $D_{shadow}^{Train}$; its corresponding prediction feature vector is given the label 1 (member) or given label 0 (nonmember) otherwise. Next, all the produced prediction vectors and labels are utilized for training the attack model $f_{attack}$. The adversary executes the attack by querying the $f$ with $x_i$ and getting its corresponding posteriors to discover whether $x_i$ is present in $D_{target}^{Train}$ (training data of target model $f$). Same as the shadow model, she obtains the three largest posteriors produced against $x_i$ and infuses them into $f_{attack}$ to retrieve its prediction as member or nonmember.

It is worth noting that here only one shadow model and one attack model are used to perform the attack, whereas MIA performed by *Rahman et al. (2018)* uses multiple shadow models and attack models typically one for each class. Thus, the approach followed in our work reduced the attack cost to a greater extent.

## Attack model without a shadow model training

In the setting of MIA without a shadow model, the adversary works without any shadow model for the attack implementation. We implement the attack model as an unsupervised classifier, for which there is no need to conduct any shadow model training. In such a situation, an adversary can depend solely on the target model's output prediction vectors $V(x_i)$ generated on target data sample $x_i$ (*Salem et al., 2019*). A similar attack designed by *Yeom et al. (2018)*, requires the target data sample's class label to effectively implement MIA making which is challenging in sensitive cases, *e.g.*, biomedical settings (*Berrang et al., 2018*). However, the attack model proposed by *Salem et al. (2019)* encompasses a broader range of scenarios.

Firstly, the adversary acquires the prediction vector $V(x_i)$ for a specific data sample $x_i$. Next, she retrieves the top posterior confidence score, compares it with a certain preset threshold, and checks whether this highest score is above it. Data sample $x_i$ is predicted as a member of the training set if the answer is yes and *vice versa*. The purpose of picking the maximum posterior score as the feature is that the DL model is more confident about predicting already seen samples (it was trained on) than the unseen samples. The reason is that, for some data examples, one posterior is exceptionally high than others, *i.e.*, the data

sample belongs to the training set of the target model. Concludingly, there is a significant difference between the highest confidence score and the remaining ones for a member data sample than a nonmember data sample. Moreover, the confidence score of a member data sample is much higher than a nonmember.

The adversary can choose a suitable threshold value for the implantation of MIA that suits well in her particular scenario, as used by various ML applications (*Zhang et al., 2017*; *Backes et al., 2017*). For example, if the prediction precision is essential she can select a relatively high threshold value. A relatively low threshold value is recommended if the adversary focuses on prediction recall (*Salem et al., 2019*).

## Differentially private deep learning model

We evaluate the impact of MIA on the DPDL model, *i.e.*, DP-BCD, since it is an efficient algorithm having appealing properties of less privacy cost consumption, speedy convergence, and prediction results with high accuracy. This particular DPDL model comprises two major segments: developing a differentially private version of the BCD algorithm, named DP-BCD, and performing a privacy analysis. The details of each of these is discussed in the following subsections.

### *Differentially private block coordinate descent*

Generally, DP can be implanted in the DL model before, after, or during training. In literature, it is found that the most suitable place to inject noise is during training. Therefore, *Riaz et al. (2023)* implement DP during model training to launch the privacy-preserved version of the BCD algorithm known as Differentially Private Block Coordinate Descent (DP-BCD). The model employs Gaussian noise instead of Laplace noise to achieve DP. The noise is drawn according to the bounded sensitivity of each sample. BCD algorithm breaks the working of deep learning problems into multi-block variables. During the forward pass, it computes the model's output at the output layer and compares it to the actual values to optimize the objective function. The block variables $(\mathcal{W}_j, \mathcal{U}_j, \mathcal{V}_j)$ also get their calculated values in forward pass. Afterward, the block variables are updated following a cyclic pattern for updating process, *i.e.*, except for the block variable updating currently, the remaining block variables keep their previous updates. This process iterates up to K epochs in backward order from the output to the input layer to get an optimal value of the objective function $\mathcal{L}$. However, in DP-BCD, before the update step, the weight block variable $\mathcal{W}_j^k$ is scaled by scaling factor C, thereby limiting the sensitivity of block variables to make sure that all samples of the training data uniformly affect the learned parameters. It helps to avoid overfitting and speedy convergence of the model. Finally, the sensitivity-dependent noise is added to the $\mathcal{W}_j^k$ which is propagated to other block variables automatically, thereby making all block variables differentially private. On this wise, DP is achieved without jeopardizing their utility and accuracy. The flow of the DP-BCD algorithm, and MIA implementation on the DP-BCD mechanism, is pictorially depicted in Fig. 2. For a detailed understanding, one may consult Algorithm 1 of *Riaz et al. (2023)* work.

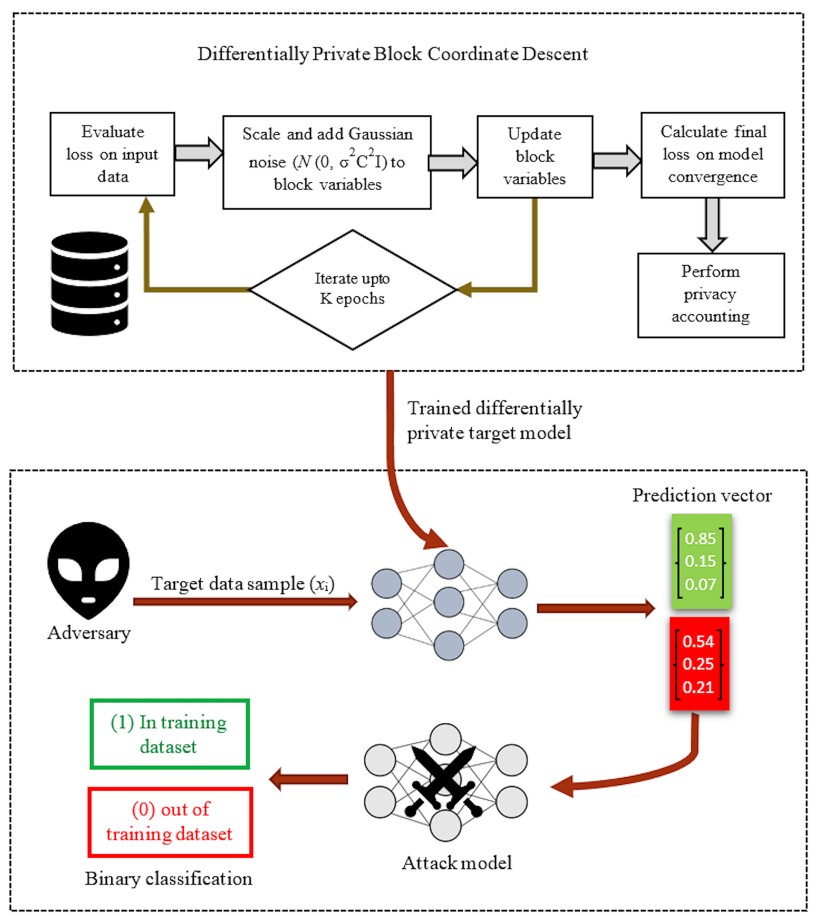

**Figure 2 Membership inference attack on DP-BCD mechanism.**

### Privacy analysis

Following the typical BCD training process, DP-BCD also requires updating the block variables repeatedly by accessing training data multiple times until the model converges to an optimum value. Nevertheless, iterating over the training data, again and again, breaches privacy and becomes the source of information leakage about the training data. Thus, it provokes some privacy cost from the total privacy budget ε. It calls for keeping track of privacy loss during the training of the model to calculate cumulative privacy cost to keep the privacy loss within the bearable budget limit. Accumulating the privacy cost acquired by each access to the model's training data is called privacy accounting and was devised by *McSherry (2009)*. In differentially private settings, the mechanism M iterates for K epochs, thus, consists of $M_K$ individual mechanisms. Each of these mechanisms has its privacy guarantee incurring a privacy cost following the basic composition theorem (*Dwork et al., 2006a*; *Dwork & Lei, 2009*) and subsequently its advanced, refined versions (*Dwork & Rothblum, 2016*; *Kairouz, Oh & Viswanath, 2015*). We follow the methods adopted by *Riaz et al. (2023)* and performed privacy accounting using strong composition (*Dwork, Rothblum & Vadhan, 2010*) contained in the existing composition theorems for DP.

However, it gives loose bounds on privacy expenditure, thereby exhausting the modest privacy budget rapidly in a few epochs, which may cause problems for models with deep layers converging after a large number of iterations. In our case, since the model converges in very early epochs, thus, consumes less privacy cost while providing acceptable utility.

To cover up the discrepancies of the strong composition theorem, *Abadi et al. (2016)* introduced a state-of-the-art technique for privacy accounting called moments accountant. It considers the tail bounds of noise distribution under consideration and provide a tighter bound on privacy spending. This capability makes moments accountant $O(r\varepsilon\sqrt{K}, \delta)$-DP by selecting an appropriate noise scale and scaling factor C having noise magnitude $\sigma$. Here, r is the per-layer ratio, and K is the number of epochs. It saves the $\sqrt{log(1/\delta)}$ in $\varepsilon$ part and Kr in $\delta$ part in case of strong composition which was formerly known for providing the best bounds on privacy accounting. The moments accountant uses log moments of $M_K$ deriving much tighter bounds on total privacy loss. As a result, it enables DP-BCD to access training data as many times as required by the training process for getting noticeable accuracy.

# EXPERIMENTAL EVALUATION AND RESULTS

In this section, we present the results of MIA implementation on the DP-BCD mechanism. Various experiments with different settings of MIA are performed to evaluate the validity of the proposed approach and to provide a comparison with existing approaches. We use three benchmark datasets for experiment evaluation, the MNIST digit recognition, Breast Cancer, and Purchase100 datasets. We selected benchmark images and numerical datasets for classification tasks in deep learning. Moreover, it practically proves that the applicability of MIA is not discriminated toward any specific dataset or model type, as exhibited by *Shokri et al. (2017)*, *Salem et al. (2019)*.

## Performance metrics used to evaluate MIA on DP-BCD

For the evaluation, we make a dataset by retrieving samples $(x_i, y_i)$ from both the training and testing datasets of the DP-BCD model with an equal ratio of 50% likelihood. A good privacy-preserving model learns from the training data but does not memorize it. This article performs empirical tests for measuring potential memorization. We launch an MIA against the target DP-BCD model to evaluate its privacy protection performance. Technically, the experiments build MIA classifiers that infer whether a particular sample is present in the training set or not. The more accurate such attack is, the more memorization is present, and thus the less privacy-preserving the model is. This privacy vulnerability (or memorization potential) is measured *via* the area under the ROC-curve (AUC) or *via* $max\{|fpr - tpr|\}$ (advantage) of the attack classifier. These measures are very closely related. Moreover, the precision and recall metrics are also calculated. The detail of these metrics is given below.

**Precision:** Precision is calculated as the true positives divided by the number of total positive (true positives + false positives) predictions. In the case of MIA, it is the proportion of training samples determined as members that are indeed present in the

training dataset of the target model. In addition, the random guessing strategy also yields 50% as the baseline precision value. The formula to calculate precision is as under:

$$Precision = \frac{TP}{TP + FP} \tag{4}$$

**Recall:** The recall is calculated as true positive divided by the true positive and false negative. In the case of MIA, it is the proportion of samples that are correctly predicted as members. Based on our threshold inference strategy, it represents the MIA's ability to predict that a training sample has a greater (or equal) prediction confidence value than the predetermined threshold. Recall is calculated using the formula given in Eq. (5).

$$Recall = \frac{TP}{TP + FN} \tag{5}$$

**Area under the ROC curve (AUC):** AUC curve is a performance measurement for classification problems at various threshold settings. ROC is a probability curve, and AUC represents the degree or measure of separability. It tells how much the model is capable of distinguishing between classes. The higher the AUC, the more accurate the model is, *i.e.*, accurately predicting the class of the data sample. In our case, members as members (class 1) and nonmembers as nonmembers (class 0). ROC curve represents the accuracy of the attacker at distinguishing between members and nonmembers.

The ROC curve is plotted with true positive rate (TPR) against the false positive rate (FPR), with TPR on the y-axis and FPR on the x-axis. Where TPR and FPR are calculated as follows:

$$TPR/Recall = \frac{TP}{TP + FN} \tag{6}$$

$$FPR = \frac{FP}{TN + FP} \tag{7}$$

**Attacker advantage:** Given a model, an attacker advantage calculates the membership attacker's (or adversary's) advantage. It measures the maximum advantage over all available classifier thresholds that characterize how well an adversary can distinguish between members and nonmembers and returns a single float number with membership attacker's advantage. Equation (8) calculates the advantage measure.

$$Attacker's\ advantage = TPR - FPR \tag{8}$$

## Attack implementation on the MNIST dataset

In this section, we discuss the MNIST dataset in detail and the results obtained by performing MIA with both settings with and without the trained shadow model on the MNIST dataset. The success of MIA is determined for different slices of data given as under:

- Entire dataset: one of the slices is the entire dataset.

- Individual classes of the dataset: one slice per class result.
- Classification correctness: slicing dataset according to the classification correctness of data samples, *i.e.*,

  – Correctly classified true
  – Correctly classified false (misclassified data samples)

### MNIST

MNIST is a standard handwritten digit recognition dataset comprising grey-level images of handwritten digits ranging from 0 to 9, with each image having a size of $28 \times 28$ pixels (*Lecun et al., 1998*). Providing an input image to the DPDL model, the classification task performed on the MNIST dataset predicts the digit written in the input image. The MNIST dataset size is 70,000 image samples with 60,000 training and 10,000 testing samples.

For MIA settings with shadow model training, we combine the training and testing samples to create a unified dataset. Then we randomly divide it into two equal halves, each consisting of 35,000 samples. The two divided datasets, namely, $D_{target}$ and $D_{Shadow}$ are used for the training and testing of the target model and shadow model respectively. For target model training, the halved dataset $D_{target}$ is further divided into $D_{target}^{Train}$ and $D_{target}^{Out}$, where $D_{target}^{Train}$ is used for target model training and $D_{target}^{Out}$ samples are supposed as nonmembers samples. Moreover, the other half $D_{shadow}$ is used for the shadow model by again splitting into two halves, $D_{shadow}^{Train}$ and $D_{shadow}^{Out}$. $D_{shadow}^{Train}$ is used for the training of shadow model and $D_{shadow}^{Out}$ is used for testing.

For an attack model without shadow model training, there is no need for a dataset for the shadow model. Therefore, we combine training and test datasets to produce a single dataset, randomly split into two equal datasets, each consisting of an equal number of samples. One-half of the dataset is used to train the target model, and the other half is used as nonmembers.

### Target model training

The target model architecture of the MNIST consists of a single hidden layer and an output layer. The only hidden layer contains 2,000 units replacing the requirement of more hidden layers, the output layer has 10 classes for 10 (0–9) digits of the MNIST dataset, the activation function ReLU is used at all layers, and Mean Squared Error (MSE) is employed as a loss function. The model comprises one hidden layer with an increased number of neuron units, and exhibits far better efficiency in terms of reduced training time (converges in early epochs) and increased model accuracy. The whole training data set is passed through the model in one batch at a time, opposing the batching method used in DP-SGD training. The model uses the default values of hyperparameters, *i.e.*, 1, 1, and 5 for $\alpha$, $\Upsilon$, and $\gamma$ respectively. For DP implementation, Gaussian noise is injected into the scaled block weight variable whose scaling in done at a scaling factor of C = 0.01. The parameters $\sigma$, r = U/N, and K contributed to the computation of total privacy cost $(\varepsilon, \delta)$.

Moreover, the target model is trained similarly for both settings of MIA, *i.e.*, with shadow model, and without shadow model. Further, the non-private model for the MNIST dataset consists of the same architecture as for the private target model.

### Training shadow model

Shadow model training is required to perform MIA with shadow model settings. Since the target DPDL model is a public model providing white box access to it. Therefore, the adversary follows the assumption of knowing the target model type and architecture; thus, the architecture of the shadow model $f_{shadow}$ is identical to its corresponding target model $f$. In an attempt to make MIA more successful, the shadow model is trained in the same settings as the target model is trained.

### Training attack model

Similarly, as with the shadow model, supervised attack model training is required to perform MIA with shadow model settings. As discussed earlier, we train a binary attack model on predictions produced by the shadow model $f_{shadow}$, which is a separate shadow model in the case with shadow model settings. Whereas, the target model $f_{target}$ itself is used as a type of $f_{shadow}$ in the case when the attack model is trained without shadow model training settings. Given an input data sample, the output of the binary attack model $f_{attack}$ is in the form of probabilities over two output classes, *i.e.*, "in" and "out," which represents the membership or non-membership of that particular data sample in the target model's training dataset, respectively. We build a simple, fully connected neural network comprising one hidden layer that contains 64 neuron units. ReLU is used as an activation function for the hidden layer, and at the output layer, SoftMax with two outputs is used to produce predictions for the "in" and "out" membership.

### Varying the privacy parameter ε

We track the impact of each setting type of MIA against two differentially private models with different noise levels, *i.e.*, heavy noise and moderate noise providing privacy guarantee of $(\varepsilon = 0.5, \delta = 10^{-3})$ and $(\varepsilon = 1, \delta = 10^{-4})$, respectively. Moreover, to gain further insight, we also track the impact of MIA on a totally non-private base model having the same network architecture as the corresponding DPDL model for each dataset.

### MIA results for trained shadow model settings

For trained shadow model MIA settings, we execute MIA against the previously trained DPDL model and evaluate its success for different noise levels. To achieve this, we calculate the AUC value as an attack accuracy evaluation metric of private models for heavy $(\varepsilon = 0.5, \delta = 10^{-3})$ and moderate noise $(\varepsilon = 1, \delta = 10^{-4})$ levels. AUC represents the relationship between TPR and FPR (*Fredrikson et al., 2014*; *Backes et al., 2017*; *Pyrgelis, Troncoso & De Cristofaro, 2017*; *Pang & Zhang, 2017*; *Zhang et al., 2018*). An AUC with a value of 0.5 is considered a random guess; however, higher than 0.5 values, on the contrary, indicate potential privacy issues. In our case, the attack performed for heavy and moderate noise resulted in an AUC value close to 0.5 (in some cases even less than 0.5), evident from Figs. 3 and 4 for heavy and moderate noise, respectively. It reveals that the attack cannot

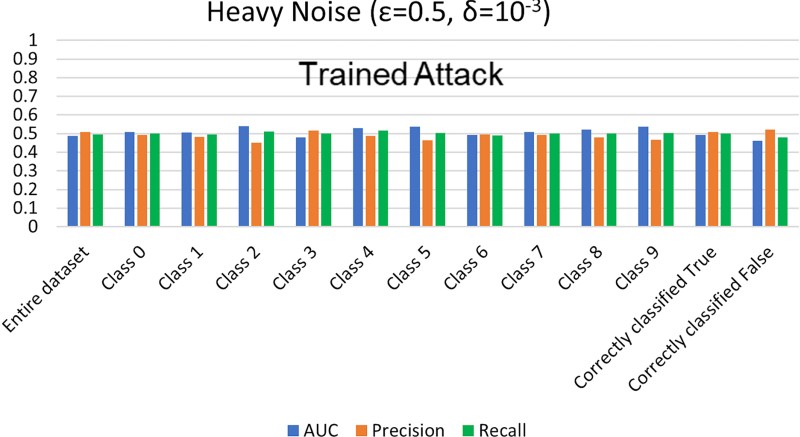

**Figure 3 Trained attack results on AUC, precision, and recall calculated for heavy noise ($\varepsilon = 0.5, \delta = 10^{-3}$).**

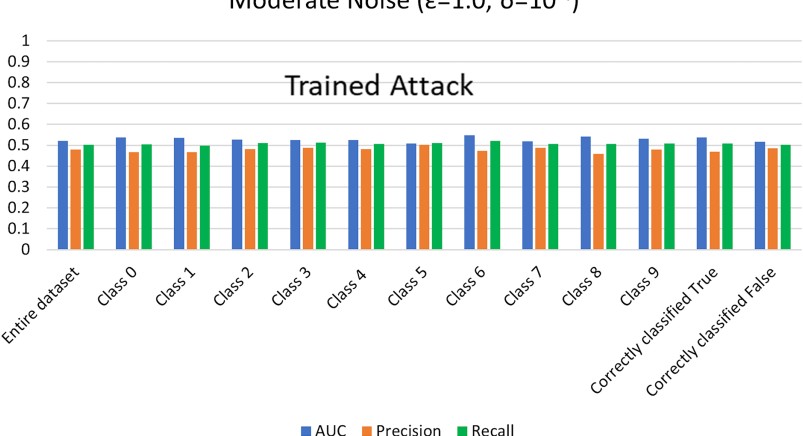

**Figure 4 Trained attack results on AUC, precision, and recall calculated for moderate noise ($\varepsilon = 1, \delta = 10^{-4}$).**

successfully cross the privacy boundaries of the DPDL model to identify its training samples. It proves that the DP-BCD preserves the model's privacy and safeguards it against inference attacks.

Besides AUC, we also calculate several other metrics, most notably precision, recall, and attacker advantage for the membership inference classifier. Tables 1 and 2 describe the detailed privacy report for each class regarding accuracy in terms of AUC calculated, precision, recall, and attacker advantage for heavy and moderate noise, respectively. In our case, the precision answers how many of the samples predicted as members are actually members, *i.e.*, the ratio of the correctly predicted members by the attack model to all predicted with member labels. Whereas recall is the number of samples correctly predicted as members out of actual members of the training dataset of the target model, *i.e.*, the ratio of the correctly predicted members to all who are members in reality. Undoubtedly, the results indicate that the attack success against the target DPDL models for both noise levels

**Table 1** MIA result summary of trained attack on MNIST dataset for heavy noise level.

**Heavy noise** ($\varepsilon = 0.5, \delta = 10^{-3}$)
**Attack type = trained attack**

| Slice feature/ Slice value | Attacker advantage | AUC | Precision | Recall | F1 score |
|---|---|---|---|---|---|
| Entire dataset | 0.21 | 0.49 | 0.51 | 0.49 | 0.50 |
| Class 0 | 0.04 | 0.51 | 0.49 | 0.50 | 0.50 |
| Class 1 | 0.22 | 0.51 | 0.48 | 0.50 | 0.49 |
| Class 2 | 0.10 | 0.54 | 0.45 | 0.51 | 0.48 |
| Class 3 | 0.23 | 0.48 | 0.52 | 0.50 | 0.51 |
| Class 4 | 0.23 | 0.53 | 0.49 | 0.52 | 0.50 |
| Class 5 | 0.08 | 0.54 | 0.46 | 0.50 | 0.48 |
| Class 6 | 0.14 | 0.49 | 0.49 | 0.49 | 0.49 |
| Class 7 | 0.22 | 0.51 | 0.49 | 0.50 | 0.50 |
| Class 8 | 0.18 | 0.52 | 0.48 | 0.50 | 0.49 |
| Class 9 | 0.11 | 0.54 | 0.47 | 0.50 | 0.48 |
| Correctly classified True | 0.05 | 0.49 | 0.51 | 0.50 | 0.50 |
| Correctly classified False | 0.19 | 0.46 | 0.52 | 0.48 | 0.50 |

**Table 2** MIA result summary of trained attack on MNIST dataset for moderate noise level.

**Moderate noise** ($\varepsilon = 1.0, \delta = 10^{-4}$)
**Attack type = trained attack**

| Slice feature/ Slice value | Attacker advantage | AUC | Precision | Recall | F1 score |
|---|---|---|---|---|---|
| Entire dataset | 0.14 | 0.52 | 0.48 | 0.50 | 0.49 |
| Class 0 | 0.09 | 0.54 | 0.47 | 0.51 | 0.49 |
| Class 1 | 0.24 | 0.53 | 0.47 | 0.50 | 0.48 |
| Class 2 | 0.22 | 0.53 | 0.48 | 0.51 | 0.49 |
| Class 3 | 0.20 | 0.53 | 0.49 | 0.51 | 0.50 |
| Class 4 | 0.16 | 0.53 | 0.48 | 0.51 | 0.49 |
| Class 5 | 0.06 | 0.51 | 0.50 | 0.51 | 0.51 |
| Class 6 | 0.24 | 0.55 | 0.47 | 0.52 | 0.50 |
| Class 7 | 0.05 | 0.52 | 0.49 | 0.51 | 0.50 |
| Class 8 | 0.20 | 0.54 | 0.46 | 0.51 | 0.48 |
| Class 9 | 0.08 | 0.53 | 0.48 | 0.51 | 0.49 |
| Correctly classified True | 0.22 | 0.54 | 0.47 | 0.51 | 0.49 |
| Correctly classified False | 0.06 | 0.52 | 0.49 | 0.50 | 0.49 |

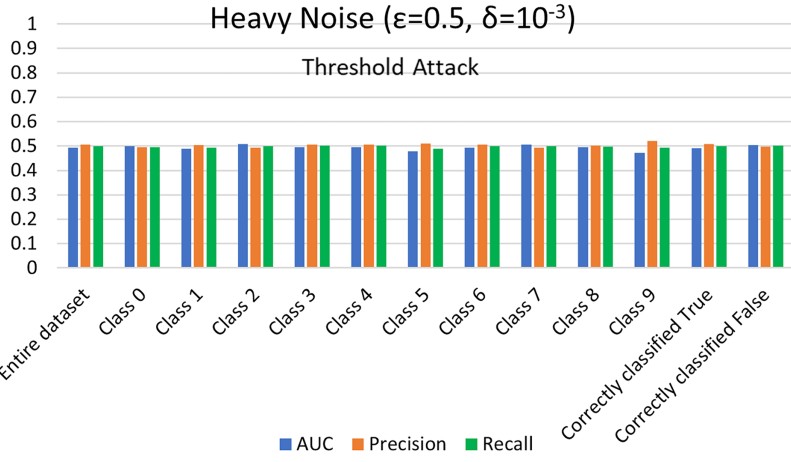

**Figure 5 Threshold attack results on AUC, precision, and recall calculated for heavy noise ($\varepsilon = 0.5, \delta = 10^{-3}$).**

is not significant and indicate the success is like a random guess accuracy almost for all of the classes. Therefore, these results prove the fulfillment of DPDL's promise to guard against a knowledgeable adversary having access to the training mechanism and the model architecture.

### MIA results without trained shadow model settings

For the MIA setting without a shadow model training, we perform the attack and assess the success of maximal prediction vector outcome on distinguishing members and nonmembers in the presence of a preset threshold. Usually, we choose a list of threshold values from 0.5 (uncertain of training or test) to 1 (100% certain of training) to compute corresponding attack evaluation metrics. The membership probability (from 0 to 1) represents each sample's probability of being in the training set. Again, the success of the attack is measured in the form of attacker advantage, AUC, precision, and recall for heavy noise ($\varepsilon = 0.5, \delta = 10^{-3}$) and moderate noise ($\varepsilon = 1, \delta = 10^{-4}$) as depicted in Figs. 5 and 6, respectively. Moreover, the detailed privacy report is described in Tables 3 and 4 for heavy and moderate noise, respectively. For a specified threshold value (0.5 in our case), we count how many training and test samples have membership probabilities larger than the threshold to compute precision and recall values. We skip the threshold value if it exceeds every sample's membership probability.

It is important to note that MIA results are almost like those obtained in the presence of a trained shadow model. It demonstrates that such a simple attack is as effective as a sophisticated attack performed with the aid of a trained shadow model. Therefore, it makes implementing MIA probably much easier and cheaper to check the privacy risks of DL models. All these empirical results verify that DP-BCD is efficient and robust enough to preserve the privacy of DL models compared to the state-of-the-art DP-SGD (*Abadi et al., 2016*).

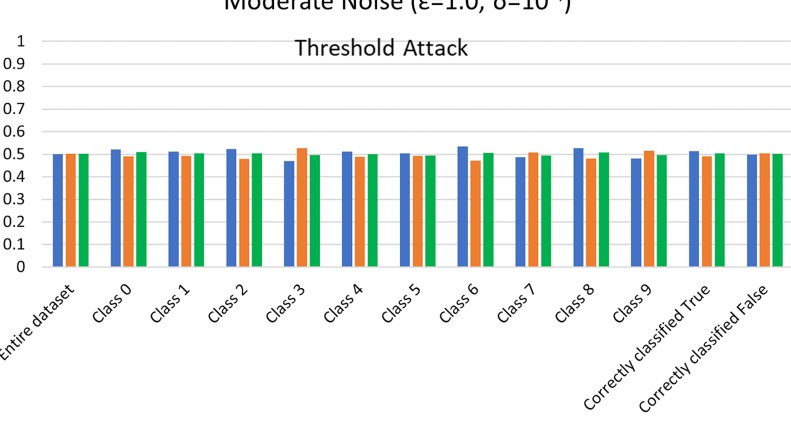

**Figure 6 Threshold attack results on AUC, precision, and recall calculated for moderate noise** $(\varepsilon = 1, \delta = 10^{-4})$.

**Table 3 MIA result summary of threshold attack on MNIST dataset for heavy noise level.**

**Heavy noise ($\varepsilon = 0.5, \delta = 10^{-3}$)**
**Attack type = threshold attack**

| Slice feature/ Slice value | Attacker advantage | AUC | Precision | Recall | F1 score |
|---|---|---|---|---|---|
| Entire dataset | 0.02 | 0.49 | 0.51 | 0.50 | 0.50 |
| Class 0 | 0.03 | 0.50 | 0.50 | 0.50 | 0.50 |
| Class 1 | 0.03 | 0.49 | 0.50 | 0.49 | 0.50 |
| Class 2 | 0.03 | 0.51 | 0.49 | 0.50 | 0.50 |
| Class 3 | 0.02 | 0.49 | 0.51 | 0.50 | 0.50 |
| Class 4 | 0.02 | 0.50 | 0.51 | 0.50 | 0.50 |
| Class 5 | 0.06 | 0.48 | 0.51 | 0.49 | 0.50 |
| Class 6 | 0.06 | 0.49 | 0.51 | 0.50 | 0.50 |
| Class 7 | 0.02 | 0.51 | 0.49 | 0.50 | 0.50 |
| Class 8 | 0.02 | 0.50 | 0.50 | 0.50 | 0.50 |
| Class 9 | 0.06 | 0.47 | 0.52 | 0.49 | 0.51 |
| Correctly classified True | 0.02 | 0.49 | 0.51 | 0.50 | 0.50 |
| Correctly classified False | 0.03 | 0.50 | 0.50 | 0.50 | 0.50 |

### Comparison with the non-private model

To compare the privacy leaked by the non-private model, we also calculate the AUC value for the non-private model as shown in Fig. 7. Likewise, the maximum AUC value calculated for heavy noise ($\varepsilon = 0.5, \delta = 10^{-3}$) and moderate noise ($\varepsilon = 1, \delta = 10^{-4}$) is depicted in Fig. 8. The private models can keep their promise to withstand privacy attacks consuming privacy costs of ($\varepsilon = 0.5, \delta = 10^{-3}$) and ($\varepsilon = 1, \delta = 10^{-4}$), respectively. It can

**Table 4 MIA result summary of threshold attack on MNIST dataset for moderate noise level.**

**Moderate noise** ($\varepsilon = 1.0, \delta = 10^{-4}$)
**Attack type = threshold attack**

| Slice feature/ Slice value | Attacker Advantage | AUC | Precision | Recall | F1 score |
|---|---|---|---|---|---|
| Entire dataset | 0.01 | 0.50 | 0.50 | 0.50 | 0.50 |
| Class 0 | 0.05 | 0.52 | 0.49 | 0.51 | 0.50 |
| Class 1 | 0.03 | 0.51 | 0.49 | 0.50 | 0.50 |
| Class 2 | 0.05 | 0.52 | 0.48 | 0.50 | 0.49 |
| Class 3 | 0.06 | 0.47 | 0.53 | 0.50 | 0.51 |
| Class 4 | 0.04 | 0.51 | 0.49 | 0.50 | 0.49 |
| Class 5 | 0.03 | 0.50 | 0.49 | 0.50 | 0.49 |
| Class 6 | 0.07 | 0.53 | 0.47 | 0.51 | 0.49 |
| Class 7 | 0.03 | 0.49 | 0.51 | 0.49 | 0.50 |
| Class 8 | 0.06 | 0.53 | 0.48 | 0.51 | 0.49 |
| Class 9 | 0.06 | 0.48 | 0.52 | 0.50 | 0.51 |
| Correctly classified True | 0.04 | 0.51 | 0.49 | 0.50 | 0.50 |
| Correctly classified False | 0.01 | 0.50 | 0.50 | 0.50 | 0.50 |

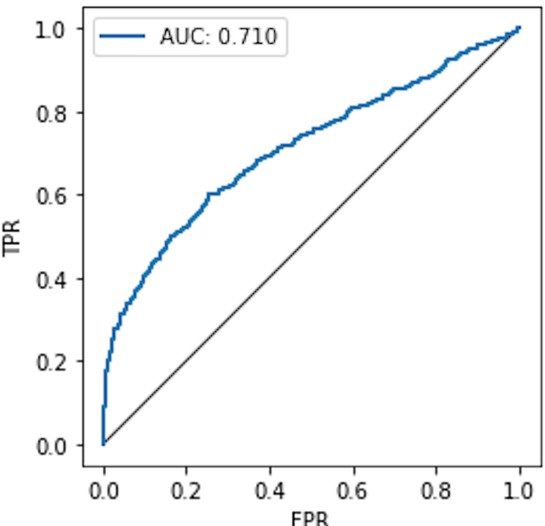

**Figure 7 AUC calculated for the non-private model.**

be seen from Fig. 8 that the maximum AUC values of private models are slightly above 0.5, whereas the non-private model exhibits vulnerability to MIA, making it almost a lousy choice. It serves as evidence that DP-BCD is robust enough to preserve the privacy of the MNIST dataset.

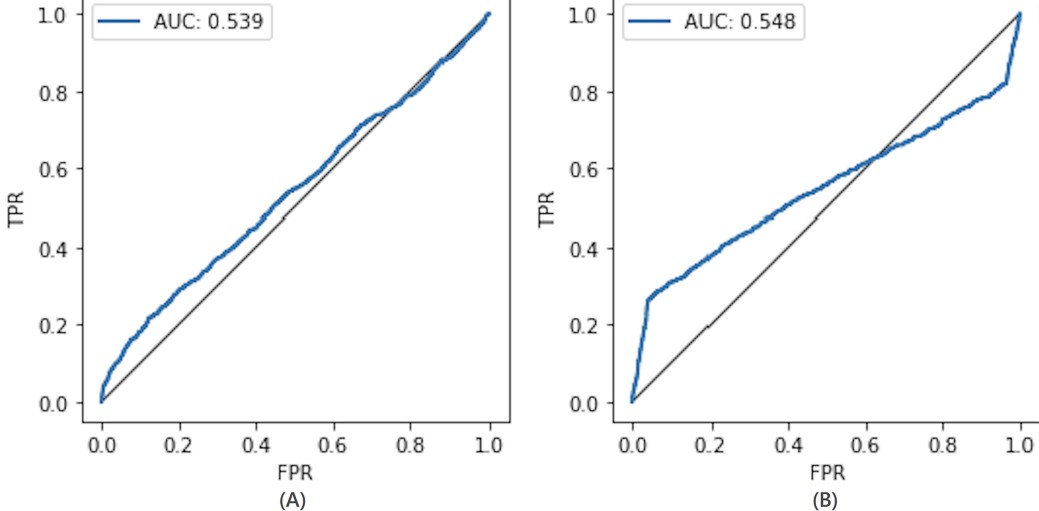

**Figure 8 Maximum AUC calculated from both trained and threshold attack for heavy noise (A) and moderate noise (B).**

**Table 5 Comparison with state-of-the-art DP-SGD.**

|  | DP-BCD | DP-SGD |
|---|---|---|
| DP privacy cost | $\varepsilon = 1.0, \delta = 10^{-4}$ | $\varepsilon = 1.0, \delta = 10^{-5}$ |
| Model utility (prediction accuracy %) | 94.6 | 75.7 |
| Attack accuracy (F1-score %) | 50.0 | 58.1 |

***Comparison with the state-of-the-art DP-SGD***

The state-of-the-art optimization algorithm used in literature for DL models is DP-SGD. *Rahman et al. (2018)* comprehensively implemented MIA on DP-SGD to verify its privacy robustness. Therefore we compare MIA results on DP-BCD with DP-SGD. We demonstrate the difference between the two by comparing privacy cost consumed, the accuracy achieved, and MIA success results in the form of an F1-score. For the same privacy budget consumption $\varepsilon = 1$, DP-BCD beats DP-SGD by providing 18.9% increase in prediction accuracy and an 8.1% decrease in MIA success accuracy in the form of F1-score as elaborated in Table 5.

## Attack implementation on Breast Cancer dataset

We also conduct experiments on the Breast Cancer dataset to perform MIA in both settings with and without shadow model training for different noise levels. MIA's success is again determined for different slices of data, *i.e.*, the entire dataset, individual classes of the dataset, by classification correctness such as correctly classified true and correctly classified false (misclassified data samples).

### Breast Cancer dataset

The Breast Cancer dataset (https://www.kaggle.com/datasets/uciml/breast-cancer-wisconsin-data) comprises 569 data examples of cancer patients. Each example instance contains 30 attributes, which are computed from a digitally created image of a fine needle aspirate (FNA) of a breast mass. These features describe the properties of the cell nuclei captured in the image. The class label is either malignant or benign, *i.e.*, the type of breast cancer. The dataset is divided into equal halves to train/test the target and shadow models.

### Target model training

A simple feedforward neural network is designed for the target model training of the Breast Cancer dataset. The model architecture consists of a single hidden layer comprising 30 neuron units and an output layer containing two classes since it is a binary classifier. A similar architecture is used for both private as well as non-private models. As far as internal parameters are concerned, for the activation of neurons, we use ReLU, and loss is calculated using mean squared error (MSE). For the private model, we apply DP using a constant noise magnitude of $\sigma = 4$ to train the model in the least time-consuming desirable privacy budget, *i.e.*, a modest cost consisting of a single-digit value. Following the original approach, the sensitivity of the block weight variable $\mathcal{W}_j^k$ is bounded using a scaling factor of $C = 0.05$. The parameters $\sigma$, r = U/N, and K contributed to the computation of total privacy cost $(\varepsilon, \delta)$. The experimented values of hyperparameters enhancing the model accuracy while preserving privacy are practically analogous to those implemented for the MNIST dataset.

Furthermore, the target model is trained in a similar fashion for both settings of MIA, *i.e.*, with shadow model, and without shadow model.

### Shadow and attack model training

The shadow and target model training is the same as for the MNIST dataset.

### MIA with trained shadow model settings

For trained shadow model MIA settings, we execute MIA against the previously trained private model for the Breast Cancer dataset and evaluate its success for different noise levels. The results in the form of AUC, precision, and recall for heavy ($\varepsilon = 0.5, \delta = 10^{-3}$) and moderate noise ($\varepsilon = 1, \delta = 10^{-4}$) levels are graphically portrayed in Figs. 9 and 10. The results depict that the AUC values of private models are about 0.5 (in some cases even less than 0.5), which advocates the competency and robustness of DP-BCD as an unbeatable private model.

Tables 6 and 7 describe the detailed privacy report for each slicing specification described above regarding accuracy in terms of attacker advantage, AUC calculated, precision, and recall. It is clear from the results that MIA does not achieve a noticeable success against the target DPDL models for both noise levels under all slicing types. Therefore, these results prove the DPDL's promise to guard against a knowledgeable adversary having access to the training mechanism and the model architecture.

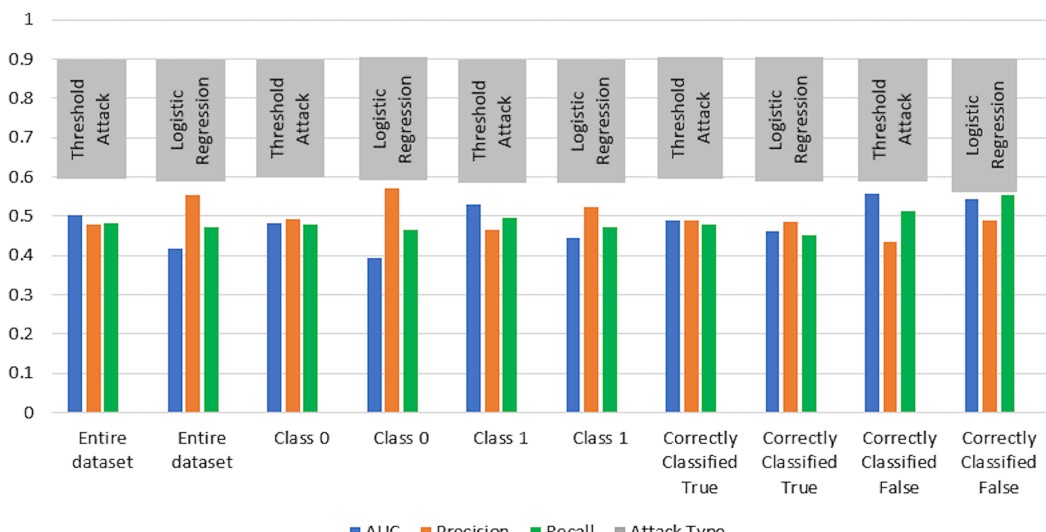

**Figure 9 Results of MIA targeting breast cancer private model calculating AUC, precision, and recall values for heavy noise ($\varepsilon = 0.5, \delta = 10^{-3}$).**

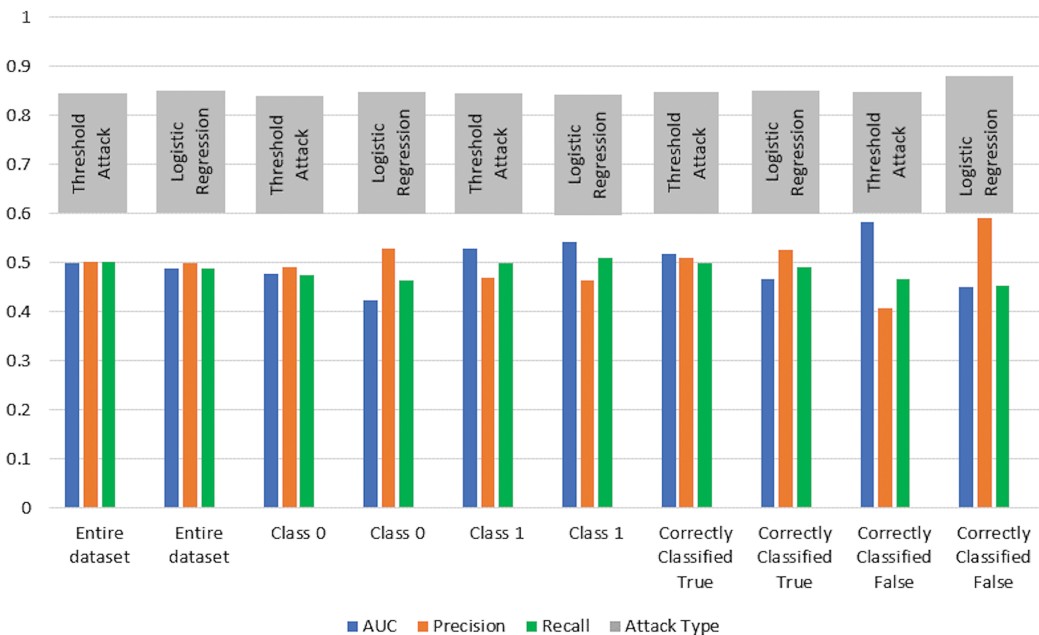

**Figure 10 Results of MIA targeting breast cancer private model calculating AUC, precision, and recall values for moderate noise ($\varepsilon = 1, \delta = 10^{-4}$).**

### MIA without trained shadow model settings

For the MIA setting without a shadow model training, we perform the attack and assess the success of the threshold attack. Similarly, the success of the attack is measured in the form of attacker advantage, AUC, precision, and recall for heavy noise ($\varepsilon = 0.5, \delta = 10^{-3}$) and

**Table 6 MIA result summary of trained attack on Breast Cancer dataset for heavy noise level.**

**Heavy noise ($\varepsilon = 0.5, \delta = 10^{-3}$)**
**Attack type = trained attack**

| Slice feature/ Slice value | Attacker advantage | AUC | Precision | Recall | F1 score |
|---|---|---|---|---|---|
| Entire dataset | 0.30 | 0.42 | 0.55 | 0.47 | 0.51 |
| Class 0 | 0.24 | 0.39 | 0.57 | 0.47 | 0.51 |
| Class 1 | 0.19 | 0.44 | 0.52 | 0.47 | 0.50 |
| Correctly classified True | 0.12 | 0.46 | 0.49 | 0.45 | 0.47 |
| Correctly classified False | 0.36 | 0.54 | 0.49 | 0.56 | 0.52 |

**Table 7 MIA result summary of trained attack on breast cancer dataset for moderate noise level.**

**Moderate noise ($\varepsilon = 1, \delta = 10^{-4}$)**
**Attack type = trained attack**

| Slice feature/ Slice value | Attacker advantage | AUC | Precision | Recall | F1 score |
|---|---|---|---|---|---|
| Entire dataset | 0.07 | 0.49 | 0.50 | 0.49 | 0.49 |
| Class 0 | 0.38 | 0.42 | 0.53 | 0.46 | 0.49 |
| Class 1 | 0.24 | 0.54 | 0.46 | 0.51 | 0.49 |
| Correctly classified True | 0.21 | 0.47 | 0.53 | 0.49 | 0.51 |
| Correctly classified False | 0.48 | 0.45 | 0.59 | 0.45 | 0.51 |

moderate noise ($\varepsilon = 1, \delta = 10^{-4}$) as illustrated in Tables 8 and 9, respectively. In the Breast Cancer dataset, again, the MIA results are almost like those obtained in the presence of a trained shadow model as depicted in Figs. 9 and 10 for heavy and moderate noise, respectively. All these empirical results verify that DP-BCD is efficient and strong enough to preserve the privacy of DL models for numeric datasets also, *i.e.*, breast cancer.

### Comparison with the non-private model

The privacy leaked by the baseline non-private model and private models (for both heavy and moderate noise) is shown in Figs. 11 and 12, respectively. The maximum AUC values calculated for the private model trained on the Breast Cancer dataset as depicted in Fig. 12 are somehow higher than that for the MNIST dataset showing the success of MIA to some extent. One of the root causes of the MIA's success is overfitting (*Shokri et al., 2017*). Overfitting is a situation where a trained model tries to memorize every sample in an attempt to fit too closely to the training data. The standard method to determine the overfitting level of a model is to find the difference between the training and test accuracies. A noticeable

**Table 8 MIA result summary of threshold attack on breast cancer dataset for heavy noise level.**

**Heavy noise ($\varepsilon = 0.5, \delta = 10^{-3}$)**
**Attack type = threshold attack**

| Slice feature/ Slice value | Attacker advantage | AUC | Precision | Recall | F1 score |
|---|---|---|---|---|---|
| Entire dataset | 0.05 | 0.50 | 0.48 | 0.48 | 0.48 |
| Class 0 | 0.09 | 0.48 | 0.49 | 0.48 | 0.49 |
| Class 1 | 0.10 | 0.53 | 0.47 | 0.50 | 0.48 |
| Correctly classified True | 0.06 | 0.49 | 0.49 | 0.48 | 0.48 |
| Correctly classified False | 0.38 | 0.56 | 0.44 | 0.52 | 0.47 |

**Table 9 MIA result summary of threshold attack on breast cancer dataset for moderate noise level.**

**Moderate noise ($\varepsilon = 1, \delta = 10^{-4}$)**
**Attack type = threshold attack**

| Slice feature/ Slice value | Attacker advantage | AUC | Precision | Recall | F1 score |
|---|---|---|---|---|---|
| Entire dataset | 0.05 | 0.50 | 0.50 | 0.50 | 0.50 |
| Class 0 | 0.09 | 0.48 | 0.49 | 0.47 | 0.48 |
| Class 1 | 0.09 | 0.53 | 0.47 | 0.50 | 0.48 |
| Correctly classified True | 0.06 | 0.52 | 0.51 | 0.50 | 0.50 |
| Correctly classified False | 0.38 | 0.58 | 0.41 | 0.47 | 0.43 |

difference indicates the overfitting of the model on the training data, and a more significant difference means a more overfitted model which exposes more information about its training data.

However, in the case of the Breast Cancer dataset, DP-BCD again keeps its promise to withstand MIA, just like in the case of MNIST. It can be observed from Fig. 12 that for both heavy ($\varepsilon = 0.5, \delta = 10^{-3}$) and moderate noise $\varepsilon = 1, \delta = 10^{-4}$, the maximum AUC values of private models are slightly above 0.5 as compared to a non-private model, which is much higher, therefore, reveals vulnerability to MIA, making it almost a bad choice. Hence, the results serve as evidence that the private models hold their promise to offer protection against membership inference for the privacy costs of ($\varepsilon = 0.5, \delta = 10^{-3}$) and ($\varepsilon = 1, \delta = 10^{-4}$), respectively.

## Attack implementation on Purchase100 dataset

To validate the performance of DP-BCD on different types of datasets, We also train the DPDL model on the Purchase100 dataset and perform MIA in both settings, with and

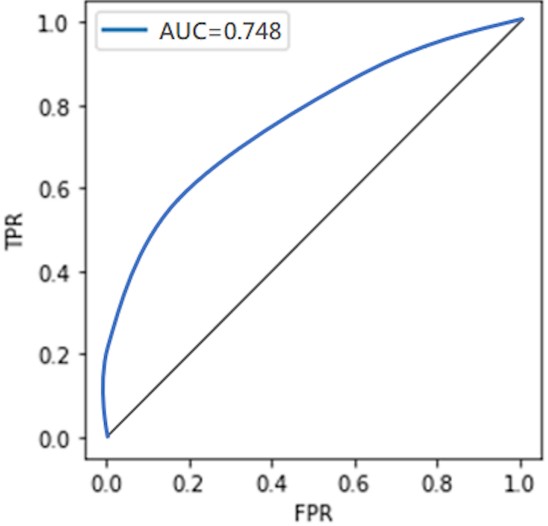

**Figure 11 AUC calculated for the non-private model.**

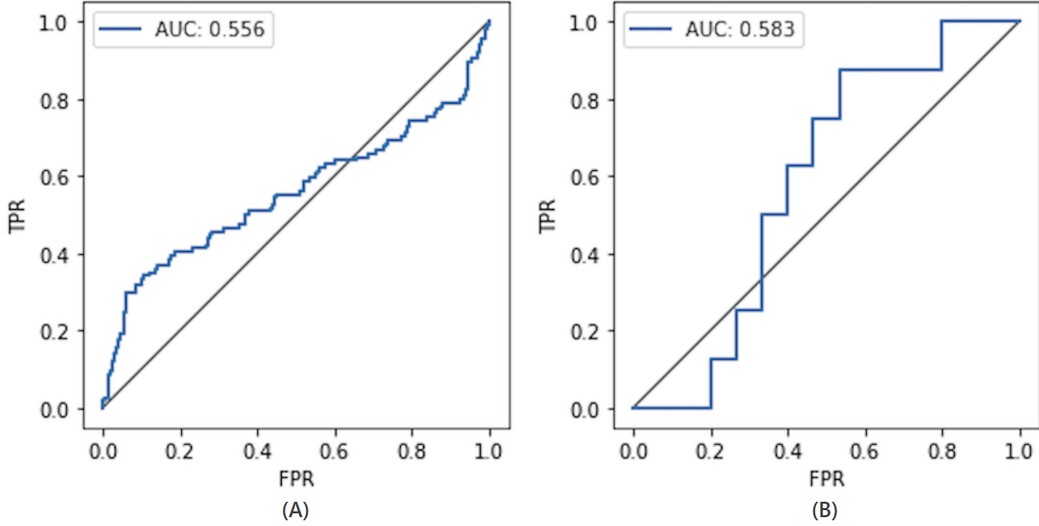

**Figure 12 Maximum AUC calculated from both trained and threshold attack for heavy noise (A) and moderate noise (B).**

without shadow model training for different noise levels. Since the Purchase100 dataset contains 100 classes, therefore, the MIA is performed on the entire dataset and by classification correctness such as, correctly classified true and correctly classified false (misclassified data samples).

### Purchase100 dataset

The Purchase100 dataset (https://www.kaggle.com/c/acquire-valued-shoppers-challenge/data) comprises customers' hopping transactions over a year. Following *Shokri et al. (2017)*, *Salem et al. (2019)*, we get a simplified version of this dataset with 197,324 records. Each record consists of 600 binary features where each feature represents a product containing a value of 0 or 1 which depicts whether the customer purchases it. For our

DPDL's classification task, we apply the K-means clustering algorithm to group the records into 100 different classes. This new version of the dataset containing 100 classes is now called Purchase100.

### Target model training

Similar to the other datasets, a simple feedforward neural network is designed for the target model training on the Purchase100 dataset. The model architecture consists of a single hidden layer comprising 1,000 neuron units and an output layer containing 100 classes. A similar architecture is used for both private as well as non-private models. The internal parameters for private and non-private models are the same as for the other datasets.

### Shadow and attack model training

The training of shadow and target models is the same as for MNIST and the Breast Cancer datasets.

### MIA with and without trained shadow model settings

We test the competency of DP-BCD based DPDL model trained on the Purchase100 dataset by evaluating MIA success for different noise levels for both settings with shadow model training and without shadow model training. Figures 13 and 14 show the results in the form of AUC, precision, and recall for heavy ($\varepsilon = 0.5, \delta = 10^{-3}$) and moderate noise ($\varepsilon = 1, \delta = 10^{-4}$) levels. The results demonstrate that the AUC values of private models are about 0.5 and even less than 0.5 in some cases for both with and without shadow model settings. These results advocate that DP-BCD keeps its promise to safeguard the private data within the Purchase100 training dataset.

Tables 10 and 11 describe the detailed privacy report regarding accuracy in terms of AUC calculated, precision, recall, and attacker advantage for heavy and moderate noise, respectively. In the case of the Purchase100 dataset, we calculate these metrics for the entire dataset and classification correctness slices. Since the dataset contains 100 classes, reporting results for every class does not seem very worthwhile. Tables 10 and 11 show that for both trained (logistic regression) and threshold attack, the adversary is unable to retrieve much information, especially in heavy noise ($\varepsilon = 0.5, \delta = 10^{-3}$). However, the results for moderate noise ($\varepsilon = 1.0, \delta = 10^{-4}$) are also very convincing since DP-BCD shows resistance to MIA for this noise level too.

### Comparison with the non-private model

To demonstrate the privacy leaked by the non-private model, the AUC value calculated for the non-private model is shown in Fig. 15. For the DPDL model, the maximum AUC value calculated for heavy and moderate noise is depicted in Figs. 16. In the case of the Purchase100 dataset, it can be observed from these figures that the maximum AUC value calculated for DP-BCD based private model is comparable to the MNIST datasets since the Purchase100 training dataset also contains a large number of samples for every class. Hence, DP-BCD retains its efficiency and privacy-preserving guarantee as in the case of MNIST and Breast Cancer datasets. Figure 16 depicts that for both heavy ($\varepsilon = 0.5, \delta = 10^{-3}$) and moderate noise $\varepsilon = 1, \delta = 10^{-4}$, the maximum AUC values of

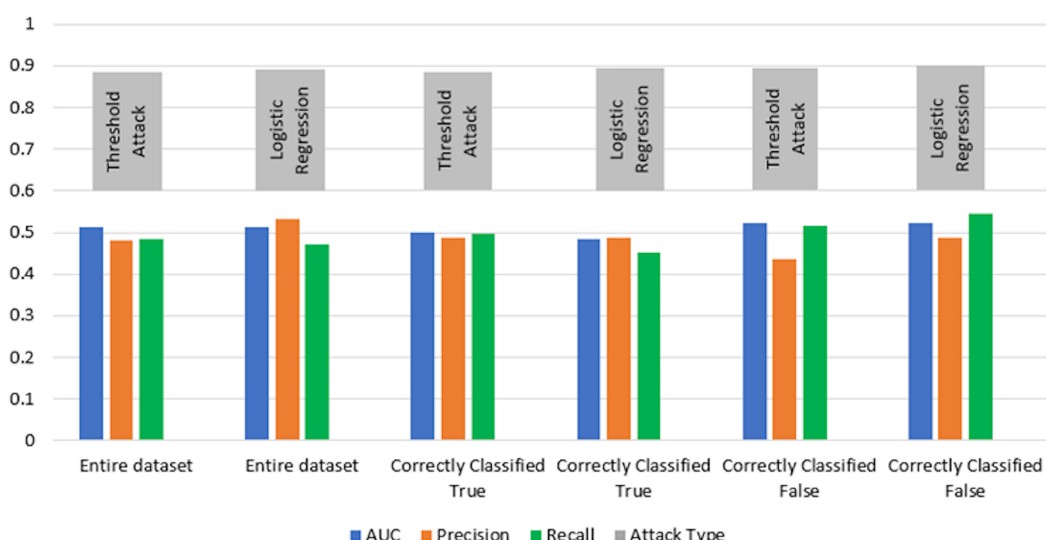

**Figure 13** Results of MIA implemented Purchase100 private model calculating AUC, precision, and recall values for heavy noise ($\varepsilon = 0.5, \delta = 10^{-3}$).

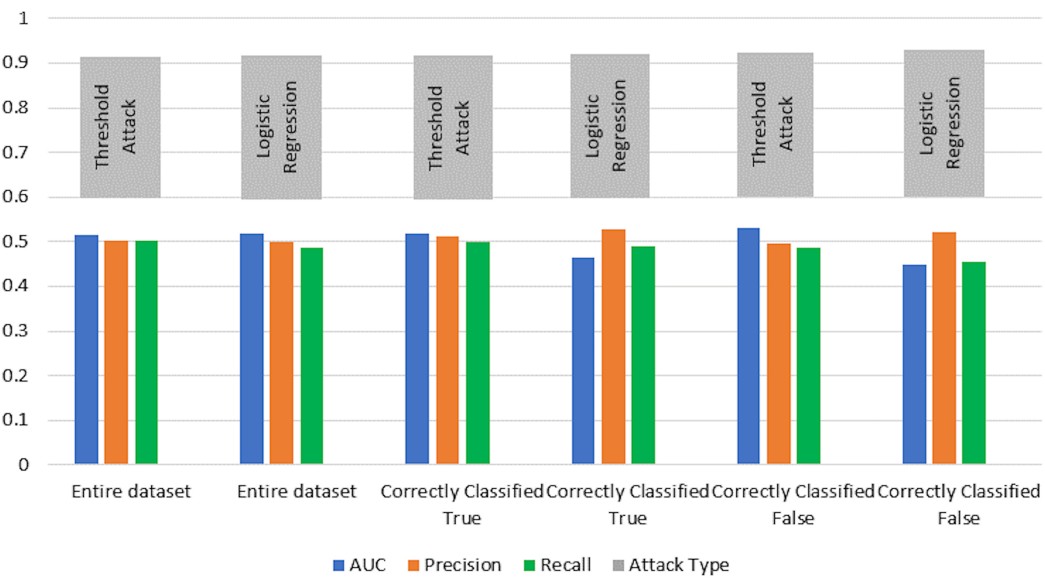

**Figure 14** Results of MIA implemented on Purchase100 private model calculating AUC, precision, and recall values for moderate noise ($\varepsilon = 1, \delta = 10^{-4}$).

private models are slightly above 0.5 as compared to a non-private model, which is considerably higher and exhibiting vulnerabilities for adversaries. Hence, DP-BCD consuming single-digit privacy cost provides the best alternative to the non-private model against MIA.

**Table 10 MIA result summary on Purchase100 dataset for heavy noise level.**

**Heavy noise ($\varepsilon = 0.5,\ \delta = 10^{-3}$)**

| Slice feature/ Slice value | Attacker advantage | AUC | Precision | Recall | F1 score | Attack type |
|---|---|---|---|---|---|---|
| Entire dataset | 0.06 | 0.51 | 0.48 | 0.48 | 0.48 | Threshold |
| Entire dataset | 0.17 | 0.51 | 0.53 | 0.47 | 0.50 | Logistic regression |
| Correctly classified true | 0.21 | 0.50 | 0.49 | 0.50 | 0.49 | Threshold |
| Correctly classified true | 0.08 | 0.48 | 0.49 | 0.45 | 0.47 | Logistic regression |
| Correctly classified false | 0.18 | 0.52 | 0.44 | 0.52 | 0.47 | Threshold |
| Correctly classified false | 0.13 | 0.52 | 0.49 | 0.55 | 0.52 | Logistic regression |

**Table 11 MIA result summary on Purchase100 dataset for moderate noise level.**

**Moderate noise ($\varepsilon = 1, \delta = 10^{-4}$)**

| Slice feature/ Slice value | Attacker advantage | AUC | Precision | Recall | F1 score | Attack type |
|---|---|---|---|---|---|---|
| Entire dataset | 0.19 | 0.51 | 0.50 | 0.50 | 0.50 | Threshold |
| Entire dataset | 0.08 | 0.52 | 0.50 | 0.49 | 0.49 | Logistic regression |
| Correctly classified true | 0.24 | 0.52 | 0.51 | 0.50 | 0.50 | Threshold |
| Correctly classified true | 0.12 | 0.47 | 0.53 | 0.49 | 0.51 | Logistic regression |
| Correctly classified false | 0.22 | 0.53 | 0.50 | 0.49 | 0.49 | Threshold |
| Correctly classified false | 0.05 | 0.45 | 0.52 | 0.45 | 0.49 | Logistic regression |

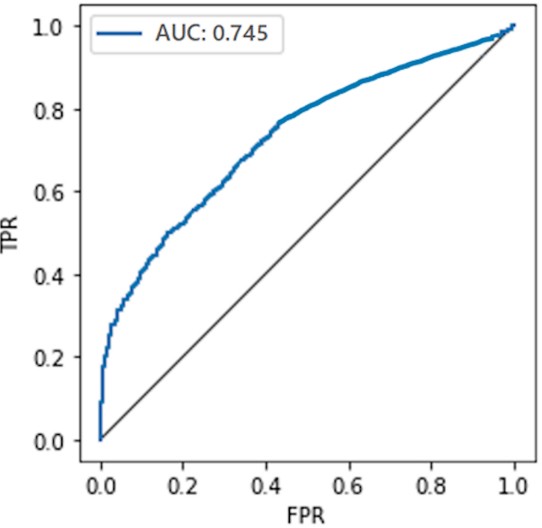

**Figure 15 AUC calculated for the non-private model.**

DP-BCD proves its promise to strike a balance between the model utility and the privacy guarantee of training data. It aims to bridge the gap between private and non-private models while providing an effective privacy guarantee of sensitive information.

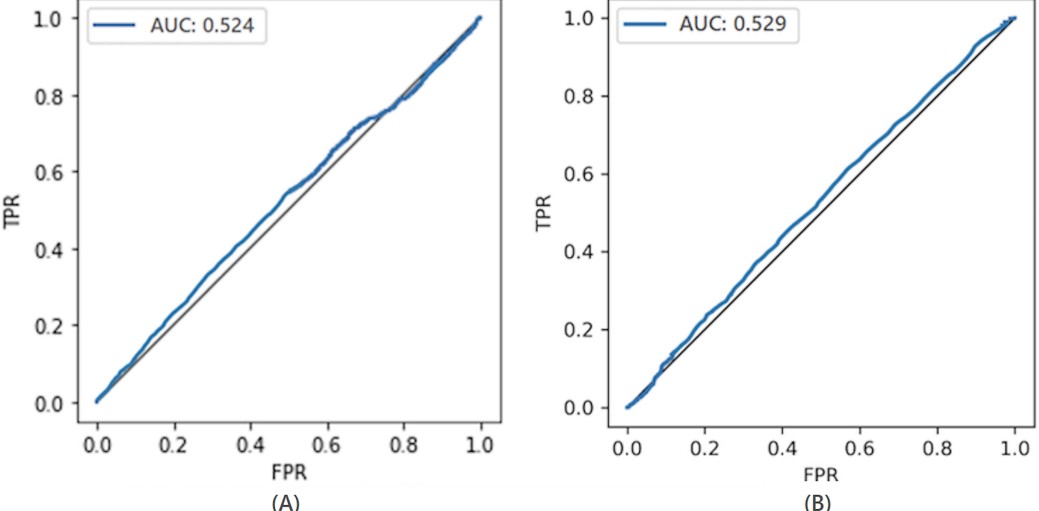

**Figure 16 Maximum AUC calculated from both trained and threshold attack for heavy noise (A) and moderate noise (B).**

**Table 12 Comparison with state-of-the-art techniques.**

| Defense technique | Model accuracy (%) | Attack performance |
|---|---|---|
| Randomized response | – | 0.57 |
| DP-SGD | – | 0.52 |
| MemGuard | 83.2 | 0.66 |
| Adversarial regularizations | 78.5 | 0.57 |
| SELENA | 79.3 | 0.54 |
| DP-BCD | 80 | 0.51 |

Experiments on benchmark real datasets demonstrate that our mechanism not only bridges the gap between private and non-private models but also prevents the disclosure of sensitive information effectively.

### Comparison with the other techniques

We compare the DP-BCD model with the state-of-the-art defense techniques proposed in the literature by comparing their robustness against MIA. *Rahimian, Orekondy & Fritz (2021)* performed MIA on DP-SGD and randomized response, whereas *Tang et al. (2022)* targeted MemGuard, adversarial regularizations, and their proposed technique self ensemble architecture (SELENA) for MIA. For DP-SGD and randomized response techniques, the authors did not provide the model's prediction accuracy and only discussed the MIA success performance. Therefore, we left model accuracy blank for these techniques in the comparison table. The MIA performance comparison of all these techniques with DP-BCD is presented in Table 12. We use attack accuracy and AUC as evaluation metrics since both metrics with 50% values mean random guess and represent an unsuccessful attack. It is evident from the table that DP-BCD is 1~15% more resistant to

MIA with noticeable accuracy as compared to other state-of-the-art techniques. Therefore, this comparison result proves that DP-BCD is strong enough to preserve the privacy of DL models against a knowledgeable adversary.

## CONCLUSION

Recently, a differentially private version of DP-BCD was developed as a substitute for state-of-the-art DP-SGD with speedy model convergence and less privacy cost consumption. However, with the development of modern attack models targeting deep models to extract sensitive information, such theoretical evaluation is not enough to prove its robustness, since it is not obvious how accurately DP-BCD trade-off utility for privacy. Therefore, in this article, we analytically evaluated the impact of a sophisticated MIA privacy attack against DP-BCD to check its practical capability. We implemented MIA with one shadow model and without shadow model training in both black box and white box settings. The shadow model presence did not give any prominent advantage to MIA, though it made the attack implementation straightforward and more efficient, which supports the results obtained for non-private deep models (*Salem et al., 2019*). The experimental results exhibit that DP-BCD keeps its promise to preserve privacy against strong adversaries while providing acceptable model utility in all settings.

Some interesting future directions suggestion include studying the effect of different neural network architectures on the success of MIA. Other than the MNIST and Breast Cancer datasets, we would like to evaluate the performance of DP-BCD on colored image datasets, such as CIFAR10. Another exciting avenue to extend the work would be to analyze the impact of attacks other than MIA (such as model inversion attack) on the DP-BCD model.

### Funding

This work was supported by the National Key Research and Development Program of China (No. 2020YFB1005804), National Natural Science Foundation of China (Nos. 62372121, 61632009), the High-Level Talents Program of Higher Education in Guangdong Province (No. 2016ZJ01), and the HEC, Faculty Development Program, Pakistan. There was no additional external funding received for this study. The funders had no role in study design, data collection and analysis, decision to publish, or preparation of the manuscript.

### Grant Disclosures

The following grant information was disclosed by the authors:
National Key Research and Development Program of China: 2020YFB1005804.
National Natural Science Foundation of China: 62372121, 61632009.
High-Level Talents Program of Higher Education in Guangdong Province: 2016ZJ01.
HEC, Faculty Development Program, Pakistan.

## Competing Interests

The authors declare that they have no competing interests.

## Author Contributions

- Shazia Riaz conceived and designed the experiments, performed the experiments, analyzed the data, performed the computation work, prepared figures and/or tables, and approved the final draft.
- Saqib Ali conceived and designed the experiments, performed the experiments, analyzed the data, performed the computation work, prepared figures and/or tables, authored or reviewed drafts of the article, and approved the final draft.
- Guojun Wang conceived and designed the experiments, authored or reviewed drafts of the article, and approved the final draft.
- Muhammad Ahsan Latif analyzed the data, authored or reviewed drafts of the article, and approved the final draft.
- Muhammad Zafar Iqbal analyzed the data, authored or reviewed drafts of the article, and approved the final draft.

## Data Availability

The data is available at the Breast Cancer Wisconsin (Diagnostic) Data Set (UCI Machine Learning): Wolberg, William, Mangasarian, Olvi, Street, Nick, and Street, W. (1995). Breast Cancer Wisconsin (Diagnostic). UCI Machine Learning Repository. https://doi.org/10.24432/C5DW2B.

The Acquire Valued Shoppers Challenge dataset is available at Kaggle: DMDave, Todd B, Will Cukierski. (2014). Acquire Valued Shoppers Challenge. Kaggle. https://kaggle.com/competitions/acquire-valued-shoppers-challenge.

The code is available at GitHub and Zenodo:

-https://github.com/ShizaMQ/Membership-inference-attack-on-differentially-private-block-coordinate-descent.

- Shazia Riaz. (2023). Membership Inference Attack on DP-BCD. Zenodo. https://doi.org/10.5281/zenodo.8251368.

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
