# Peer review of "Membership inference attack on differentially private block coordinate descent"

_PeerJ Computer Science, doi:10.7717/peerj-cs.1616_

## Round 0.1 · original submission · Major Revisions

I have received reviews of your manuscript from two scholars who are experts on the cited topic. They find the topic very interesting; however, several concerns must be addressed regarding experimental results, more datasets, and comparisons with current approaches. These issues require a major revision. Please refer to the reviewers’ comments listed at the end of this letter, and you will see that they are advising that you revise your manuscript. If you are prepared to undertake the work required, I would be pleased to reconsider my decision. Please submit a list of changes or a rebuttal against each point that is being raised when you submit your revised manuscript.

Thank you for considering PeerJ Computer Science for the publication of your research.

With kind regards,

Reviewer 1 ·

Basic reporting

• The basic reporting of the manuscript is clear and the problem statement is clearly explained

Experimental design

• The manuscript gives a good comprehensive study on Membership inference attack on DP-BCD. It would have been nice if you had highlighted experimental analysis different Deep learning models on different attacks and its solution. It is also suggested to work on the different categories of datasets.

Validity of the findings

• More comparisons on different datasets are required.

Additional comments

• More recent papers should be reviewed. There are still some more profound works that are not considered in this work.
The writing of this manuscript needs further improvements. There are some grammatical mistakes.

Cite this review as

Reviewer 2 ·

Basic reporting

Even if the aim of the project is clearly given, The references are too old. The recently proposed techniques especially deep learning based ones must be added and compared with the proposed technique.

Experimental design

Experimental results were given but, the results and comparisons with the recently proposed techniques must be added also.

Validity of the findings

The results of the other benchmark tests should also be given.

Additional comments

The paper should be carefully revised by a native English speaker or a professional language editing service to improve the grammar and readability. The results must be justified and verified by the other techniques proposed recently.

Cite this review as

---

## Round 0.2 · accepted · Accept

I am pleased to inform you that your work has now been accepted for publication in PeerJ Computer Science.

Please be advised that you are not permitted to add or remove authors or references post-acceptance, regardless of the reviewers' request(s).

Thank you for submitting your work to this journal. On behalf of the Editors of PeerJ Computer Science, we look forward to your continued contributions to the Journal.

With kind regards,

Reviewer 1 ·

Basic reporting

The authors have answered my review concern very well in the manuscript.
The manuscript has been revised as per the reviewer's suggestions and it can be considered for publication

Experimental design

The manuscript has been revised as per the reviewer's suggestions and added a subsection “Comparison with other techniques” to compare the MIA results on DP-BCD to other state-of-the-art techniques, i.e., DP-SGD, Randomized Response, MemGuard, Adversarial Regularizations, and SELENA (Self Ensemble Architecture).

Validity of the findings

-

Additional comments

The manuscript can be considered for publication

Cite this review as

Reviewer 2 ·

Basic reporting

The most of the reporting issues are fixed in the last version of the manuscript.

Experimental design

The most of the points raised by the reviewers have been addressed satisfactorily in the latest version.

Validity of the findings

Some comparisons with the recently proposed techniques are added.

Additional comments

Even the writing style still needs to be some improvements, it is acceptable.

Cite this review as